# Prevalence of Metabolic Syndrome among Early Adolescents in Khartoum State, Sudan

**DOI:** 10.3390/ijerph192214876

**Published:** 2022-11-11

**Authors:** Fatima A. Elfaki, Aziza I. G. Mukhayer, Mohamed E. Moukhyer, Rama M. Chandika, Stef P. J. Kremers

**Affiliations:** 1Department of Clinical Nutrition, Faculty of Applied Medical Sciences, Jazan University, Jazan P.O. Box 114, Saudi Arabia; 2School of Nutrition and Translation Research in Metabolism, Maastricht University, 6211 LK Maastricht, The Netherlands; 3Department of Health Education and Promotion, Maastricht University, 6211 LK Maastricht, The Netherlands; 4School of Medicine, Ahfad University for Women, Omdurman P.O. Box 167, Sudan; 5Department of Emergency Medical Services, Faculty of Applied Medical Sciences, Jazan University, Jazan P.O. Box 114, Saudi Arabia; 6Public Health Programs, School of Medicine, University of Limerick, V94 T9PX Limerick, Ireland

**Keywords:** metabolic syndrome, adolescents, prevalence, noncommunicable diseases, obesity, Sudan

## Abstract

Background: Metabolic syndrome (MetS) is rapidly increasing in prevalence with rising childhood obesity and sedentary lifestyles worldwide. The aim of this study was to estimate the prevalence of MetS and its components among Sudanese early adolescents in Khartoum State. Methods: A descriptive cross-sectional study was conducted at primary schools in Khartoum State. A questionnaire was administered to assess the sociodemographic characteristics of the participants. Anthropometric, blood pressure, and biochemical measurements were taken. Results: In total, 921 students, boys and girls aged 10–15 years old, participated in the study. The mean age of the participants was 12.59 ± 1.21 years. The overall prevalence rate of MetS was 2.3% using International Diabetes Federation (IDF) criteria. MetS was significantly more prevalent among boys than girls (3.4% vs. 1.5%). Obese adolescents had higher MetS prevalence than those who were overweight (14.9 vs. 2.8, *p* < 0.001). Conclusion: Boys had a significantly higher prevalence of metabolic syndrome than girls. Early adolescents from Sudan who are obese had more risk factors for MetS than those who are normal weight or overweight. It is important to address the causes of increased risk for MetS early in life to prevent the development of the disease in adult life.

## 1. Introduction

Noncommunicable diseases (NCDs) are the leading causes of death globally [1]. The burden of NCDs is progressively increasing. According to the World Health Organization (WHO), future projections indicate an alarming increase in the prevalence of NCDs, with four main types (including type 2 diabetes and cardiovascular diseases) causing as many as 2.4 million deaths in 2025 [2]. Sudan faces a double burden of diseases, with rising rates of communicable and noncommunicable diseases [3]. Since 1998, diabetes mellitus, cardiovascular diseases (CVD), and cancer have been among the top ten causes of hospital admissions and deaths in Sudan [3,4], as well as a double burden of malnutrition characterized by the coexistence of undernutrition along with overweight and obesity. As a major risk factor for chronic disease, metabolic syndrome (MetS) is rapidly increasing in prevalence with rising childhood and adolescent obesity and sedentary lifestyles worldwide [5,6].

The integrated epidemiological concept of MetS was created based on the observation that some metabolic risk factors occur concomitantly in patients at high risk of CVD, notably abdominal obesity, dyslipidemia (elevated triglycerides and lowered high-density lipoprotein cholesterol), high blood pressure, impaired fasting glucose, and insulin resistance [7,8]. Metabolic syndrome is a clustering of metabolic abnormalities associated with risk of coronary heart disease, stroke, and type 2 diabetes. It involves insulin resistance (IR), which is central to the pathogenesis of type 2 diabetes mellitus, visceral obesity, and systemic inflammation, as well as cardiovascular diseases. Children and adolescents with MetS have an increased risk of acquiring MetS as adults. Combined, these diseases constitute an ever-increasing burden of patient suffering and social costs [9]. Familial influences on the occurrence of MetS operate through shared environmental and genetic factors [10]. A genetic risk that accompanies family history of atherosclerotic CVD is observed for high lipid concentrations, high blood pressure, and high glucose concentration [10]. However, some authors argue that while genetic influence in individual components is possible, MetS is a combination of five components and no single genetic entity has been found to be associated with MetS [11].

Previous studies have revealed an association between obesity and the clustering of metabolic abnormalities in early life and their persistence during adulthood [12]. Thus, it has become crucial to gain a better understanding of MetS’s prevalence, pathophysiology, and risk factors and to identify strategies for managing it, especially in childhood and adolescence.

Worldwide, variations in the prevalence of MetS are observed due to using different criteria across different parts of the world [13]. Various publications have shown prevalence rates ranging from 0.2% to 38.9% among young population groups [14]. A systematic review among children reported that the total median prevalence of MetS was 3.3% (range 0–19.2%). In overweight children, the median prevalence was 11.9% (range 2.8–29.3%), and in obese populations it was 29.2% (range 10–66%) [15].

The pubertal stage during early adolescence is described as a critical phase of rapid growth in a person’s life where many physiological changes occur. The multiple burdens of malnutrition, metabolic syndrome, and chronic diseases are strongly linked in early adolescence. However, there is an increasing prevalence of NCDs in low- and middle-income countries [16], which makes it necessary to study the prevalence of MetS among early adolescents.

Additionally, the aims of the Sudanese national government’s 25-year plan (2003–2027) are to reduce the burden of diseases, promote healthy lifestyles, develop and retain human resources, and introduce advanced technology, while ensuring the equity, quality, and accessibility of health services [3]. To do so, the government needs to be clear on how widespread health issues currently are in the country, especially with regard to vulnerable groups such as children and adolescents. Given the lack of sufficient published studies on the prevalence of MetS among Sudanese early adolescents, the aim of this study was to estimate the prevalence of MetS and its components among Sudanese early adolescents in Khartoum State.

## 2. Materials and Methods

### 2.1. Study Design and Population

A descriptive cross-sectional school-based study was conducted in Khartoum State during the period of 2018–2019. Khartoum is a state and also the capital of the Republic of the Sudan and hosts all the varied ethnic groups of population. The current population is diverse in terms of ethnicity, due to migration from all regions of Sudan seeking employment and improved living conditions. The two localities from Khartoum state were selected by ballot: Khartoum city from the urban area and Karari from the rural area were selected for the purposes of the present study. The study population comprises adolescents within the age group of 10–15 years from primary schools in Khartoum State. The exclusion criteria include students with type 1 diabetes or any mental or physical limitations that would impede the completion of the interview and assessments, as well as pregnant and lactating girls.

A three-stage cluster, proportionate to population size, and systematic sampling procedure was used to obtain the representative sample. By the cluster sampling method, schools were selected from each locality with the random number tables, then the required number of students from the selected schools was calculated with the proportion of school enrollment size to the total number of students in the selected locality. A complete list of all early adolescents aged 10–15 years was taken from the school authorities. A systematic random sampling procedure was used to select the desired number of students from a list. If the requisite sample of school-age children was not available in the identified cluster, the nearest school was included to complete the sample size. A total of 921 students aged 10–15 years were enrolled.

### 2.2. Data Collection

A training workshop for all data collectors was conducted by qualified trainers before data collection. This workshop covered standardized methods of taking anthropometric, blood pressure, and biochemical measurements.

The interviewers were asked to interview some students to pretest the questionnaire in the presence of the researcher and as part of the training. The teams were given a demonstration on the technique of using weighing scales, emphasizing that the scale should be adjusted to zero before each measurement with the scale screw and that heavy clothes and shoes should be removed during measurement.

Personal identification numbers were assigned to the participants to maintain anonymity. The identification numbers confirmed the participants’ consent status and linked the students to their respective schools, clinical measurements, and blood samples. Confidentiality of the identity of all the subjects was fully ensured, and only aggregate data were reported and released.

### 2.3. Study Instruments

Three types of data were collected from the study:(i)A structured closed-ended questionnaire was filled by a data collector for each student to obtain information on the student’s sex, age, and residence and his or her parents’ educational level.(ii)Physical measurements, including anthropometric (weight, height, and waist circumference) and blood pressure measurements were taken. Weight was measured using a digital scale (SECA with light clothing). To determine the participants’ height, a portable stadiometer was used while they were standing on bare feet.

The estimation of the prevalence of overweight and obesity was based on age- and sex-specific BMI cutoff points developed by the WHO [17,18]. Overweight and obesity were defined as BMI-for-age and height-for-age (*z*-score). Overweight was defined as >+1SD, whereas obesity was defined as >+2SD and underweight as <−2SD.

Waist circumference (WC) was measured in centimeters (cm) using an unstretched measuring tape to measure the midpoint between the bottom of the rib cage and the area above the tip of the iliac crest, to the nearest 0.1 cm. Children were determined as having abdominal obesity when their WC was ≥90 cm percentile.

Blood pressure was measured using an automatic digital blood pressure monitor with an adjustable arm cuff for different arm sizes. The participants were seated with their right arm resting at the level of the heart. Their blood pressure was recorded twice within an interval of 5 min to reach the relaxation and stabilization of the blood pressure, and the average value of their systolic and diastolic blood pressure from the two readings was taken.

(iii)For biochemical measurements, 5 mL of venous blood sample was obtained from the participants by a qualified lab technician using standardized tubes: lithium heparin tube for lipid profile and fluoride oxalate tube for fasting blood glucose. Before blood was withdrawn, all the participants had to fast overnight for at least 8 h. After blood collection, the samples were centrifuged (L500 Tabletop Low Speed Centrifuge) for 10 min at 3000 rpm, allowing enough time for blood clotting.

Blood samples were transferred to a laboratory at Almogran University Hospital. All quality controls were performed to ensure the accuracy of the analytical testing.

To assess triglycerides (TG) and high-density lipoprotein cholesterol (HDL-C), serum samples were estimated for TG and HDL-C using the enzymatic method automated by an A25 Analyzer.

To assess fasting blood glucose (FBG), glycemia was measured through the enzymatic glucose oxidase method using automation equipment, the A25 analyzer. Fasting blood glucose, high-density lipoprotein cholesterol, and triglycerides were analyzed by an A25 Biosystems SA Costa Brava 30, Barcelona (Spain) Analyzer using the appropriate conventional laboratory reagents and enzymatic and calorimetric techniques. High-density lipoprotein cholesterol, serum triglycerides, and fasting blood sugar were assessed by the standard enzymatic kit method using a full automation analyzer.

Cutoff points for the MetS and abnormal values for the components were taken from International Diabetes Federation (IDF) guidelines [19]. For children aged 10–16 years, metabolic syndrome was diagnosed with abdominal obesity (waist circumference ≥ 90th percentile) and the presence of two or more other clinical features (triglycerides (TG)  ≥  150 mg/dL, HDL-C  <  40 mg/dL, systolic blood pressure (BP)  ≥  130 mmHg or diastolic BP  ≥  85 mmHg, fasting plasma glucose (FG)  ≥  100 mg/dL) [19].

### 2.4. Data Analysis

Statistical analyses were performed with the IBM SPSS statistics version 22. Quantitative data were presented in the form of mean and standard deviation and categorical variables were presented in numbers and percentages. Nonparametric chi-square test was applied for assessing the gender difference in baseline characteristics (Table 1), MetS association with baseline characteristics (Table 2), and MetS association with its components (Table 3). Stepwise multivariable regression method was applied to assess the strength of association between MetS and its components (Table 4). MetS prevalence trend with its components—WC, TG, HDL, FBG, and BP—according to BMI, is shown in Figure 1. *p* value was less than 0.05.

### 2.5. Ethical Consideration

The present study was approved by the Research Ethics Committee of the Federal Ministry of Health (Ref, No. 1-12-17). Approval was also secured from the Ministry of Education, as well as from school principals at the selected schools. Students voluntarily agreed to take part in the study after being informed verbally and through informed written consent, which was obtained from the students and their parents and/or guardians. The students were told that the collected information would be kept anonymous and that they had the right to withdraw from the study at any time.

## 3. Results

### 3.1. General Background Characteristics of the Study Population

The present study was carried out on 921 early adolescents, of whom 388 (42.1%) were boys and 533 (57.9%) were girls. The mean age and standard deviation of the participants was 12.59 ± 1.21 years (12.69 ± 1.16 for boys and 12.52 ± 1.23 for girls). Approximately two-thirds of the boys (63.7%) were from urban areas, whereas in girls more than half were from rural areas (53.5%), and the difference between sexes was highly significant (*p* < 0.01). The highest number of participants (325; 35.3%) was from Grade 8, among which were 156 boys (40.2%) and 169 girls (31.7%). Regarding parents’ education, only 26 (2.8%) fathers were illiterate. Nearly half of the fathers (380; 41.2%) had university/postgraduate-level qualifications, whereas in mothers, 52 (5.6%) were illiterate and one-third (302, 32.8%) of the mothers had qualifications from the university/postgraduate level.

The overall prevalence of overweight and obesity was 11.7% and 9.4%, respectively. Overweight was significantly more prevalent among girls than boys (13.5% vs. 9.2%), whereas obesity was slightly higher in boys than girls (9.8% vs. 9.3%) (*p* = 0.004). In total, the prevalence of underweight was 16.6% (Table 1).

### 3.2. Association between Prevalence of Metabolic Syndrome and Demographic Characteristics

Table 2 depicts the association between MetS and sociodemographic characteristics. The overall prevalence of MetS was 2.3%. The prevalence rate was higher among boys than among girls, at 3.4% vs. 1.5%, respectively. This difference, however, is not statistically significant (*p* = 0.063). The prevalence of MetS from the ages 10 to 15 years ranged from 1.8% to 3.1%, respectively. Residing in urban areas was seen to have a higher prevalence of MetS as captured in the table (3.4% urban and 0.9% in rural) and it was found to be statistically significant (*p* < 0.01). Grade 8 students (12 out of 325 accounting for 3.7%) witnessed a higher prevalence of MetS in comparison with other students (Grade 5 (0.7%), Grade 6 (0.9%), and Grade 7 (2.7%)). A significantly higher prevalence of MetS (14.9%) was observed among obese early adolescents compared with underweight (0.7%), normal weight (0.7%), and overweight children (2.8%), and the association was found to be highly significant (*p* < 0.01).

### 3.3. Prevalence of Metabolic Syndrome Components

Waist circumference, as an essential component of MetS, was found in all those with metabolic syndrome. High waist circumference was found in 82 (8.9%), and 25.6% of them were diagnosed with metabolic syndrome. With respect to blood pressure, it was found that high diastolic BP (20.9% of the participants) was diagnosed in children more often than high systolic BP (4.7%). Metabolic syndrome was confirmed in 11.6% of early adolescents diagnosed to have higher BP. Fasting blood glucose estimation showed that 65.9% of the total study participants had high fasting blood glucose. Characteristically, all metabolic syndrome early adolescents were diagnosed with increased fasting blood glucose levels. In total, 6.9% of the participants had high triglycerides, which is a potential threat to the growing community. Almost half (10 out of 21 MetS early adolescents) were diagnosed as having high triglyceride content in the blood. In non-MetS early adolescents, only 1.3% were found to have high triglycerides. Protective cholesterol (HDL) was found to be low in a substantial set of participants (15%). HDL in MetS early adolescents was found to be low in 61.9% in comparison with 13.8% of non-MetS early adolescents. The value in the non-MetS early adolescents is also a major concern, as decreasing HDL can lead to MetS in the future (Table 3).

Figure 1 shows the MetS prevalence trend with its components—WC, TG, HDL, FBG, and BP—according to BMI. Top left side figure represents the MetS association with BMI and WC. A high prevalence of MetS was observed among early adolescents with high WC. Top right side figure shows an increased prevalence trend (5.9% to 41.7%) among elevated-TG early adolescents when their BMI is increasing. Bottom left side figure shows an increased prevalence trend from 4.7% to 50% among early adolescents when their BMI is increasing. An increased prevalence of MetS (5.6% to 50%) was noticed among overweight and obese early adolescents. Contrarily, in HDL normal early adolescents, the prevalence of MetS steadily increased from 0% to 8.2% with increased BMI. Bottom middle figure shows an increased trend from a 1.1% to 22.8% prevalence of MetS among high-FBG early adolescents with increased BMI. A rapid increased prevalence of MetS (4.2% to 22.8%) was noticed among overweight and obese early adolescents. Bottom left side figure shows an increased trend from a 0% to 55.6% prevalence of MetS among elevated blood pressure and obese early adolescents.

### 3.4. Association between Metabolic Syndrome and Its Components

A multivariable regression analysis with stepwise models indicates that MetS was significantly positively associated with the components high WC, high TG, low HDL, high FBG, and high HTN. (Table 4). High WC was the highest significant component for the MetS, which is shown in Model 1, standardized coefficient (β) = 0.489, and high significance (*p* < 0.001), followed by high TG, low HDL, and high FBG. High HTN had the least association with MetS.

## 4. Discussion

The current study is the first to explore the prevalence of metabolic syndrome (MetS) among Sudanese early adolescents living in Khartoum State. Given the importance of preventing and treating the underlying risk factors for MetS, the present study will help prioritize the topic of early adolescent metabolic syndrome in the Sudanese context and develop strategies for preventing the risk of developing heart disease and type 2 diabetes and reducing their social cost. Compared with the findings of previous studies conducted in Sudan, our findings yield a higher overweight status than that reported by Mukhayer in 2014 (9.1% overweight or obese) among Sudanese adolescents [20] and similar results were found with the study conducted among 10- to 18-year-olds in Sudan (19.8%) [21]. On the other hand, the prevalence rates in our study were lower than those found in another Sudanese study conducted among 6- to 12-year-olds (25.3%) in 2010 [22].

Other studies conducted in Sudan investigated the prevalence of MetS among university students. In 2015, Ahmed et al. found that the overall prevalence rate of MetS in those 18–23 years old was 7.8%, while another study showed prevalence rates of 8.4% and 7.5% [23]. However, the overall prevalence rates of MetS vary based on the criteria used. In the present study, we analyzed the prevalence of metabolic syndrome in early Sudanese adolescents aged 10–15 years old. Using the IDF criteria [24], the prevalence of MetS was 2.3%. Compared with previous studies, this study determined a prevalence that was lower than those found in most studies among similar age groups in other countries, including India, Iran, Canada, Qatar, Saudi Arabia, Iran, the United States, United Arab Emirates, and Northern Mexico [12,24,25,26,27,28,29,30,31,32,33,34] and higher than those found in studies in China, Northern Mexico, Middle East and North Africa, and Jordan [10,34,35,36]. In the present study, the prevalence was similar to a study conducted among schoolchildren 10–19 years old in Turkey [37].

Our findings showed that the prevalence of MetS was higher in boys than in girls. These findings are in line with the results of studies conducted in the United States, United Arab Emirates, South Africa, and Saudi Arabia [32,33,38,39]. Increased prevalence in boys may be related to differences in sex hormones, including testosterone and sex hormone-binding globulin (SHBG), which are substantially produced throughout puberty, as well as growth differences between girls and boys during puberty [40,41]. Other studies conducted among university students in Sudan [24] and Saudi Arabia [9] found that the prevalence of MetS was higher among girls than boys.

In this study, the prevalence of metabolic syndrome was found to be higher in urban areas than in rural areas, in line with the results of previous Saudi Arabian and Sudanese studies [9,24]. These studies reported that their findings could be attributed to urbanization and adoption of Western lifestyles.

It is well-documented that childhood obesity is the main risk factor associated with the development of MetS [25,26,27]. In our study, a higher prevalence of metabolic syndrome and individual risk factors was observed among obese early adolescents (14.9%) than among overweight ones (2.8%). The rate of MetS increased progressively with higher body mass index. The prevalence of MetS has been reported to be higher in obese early adolescents (56%) [42]. This is consistent with the results of Simunovic et al. [43], who showed that the risk of MetS increases with obesity. Compared to our findings, there were lower prevalence rates of MetS (5.3%) in other groups of obese children in a previous study by [44]. Screening for MetS component is important in early identification and subsequent intervention, as even the slightest increase in MetS components poses a future risk of cardiovascular diseases [45]. In the present study, it was noticed that MetS early adolescents (aged 10–15 years) had high WC, high triglycerides, high blood pressure, low HDL, and high glucose respectively. These factors are shown to contribute to the increase in cardiometabolic disorders [45,46,47].

Elevated triglycerides were the most common component in MetS after high waist circumference, according to the findings of the current study. This result was similar to those of previous studies in the same age group in other countries [30,32,48]. Other studies showed HDL as the most common component [33,35]. Elevated blood pressure appeared to be the least prevalent component.

The main strength of our study was its large sample size, and efforts were made to represent a reliable sample for early Sudanese adolescents in Khartoum State, with a good response rate. In addition, all the components of MetS were assessed objectively using validated measures. One of the limitations of the current study was its cross-sectional design, which did not allow for the establishment of temporal causal relationships.

## 5. Conclusions

Among Sudanese early adolescents, boys had a significantly higher prevalence of metabolic syndrome than girls. Obese adolescents exhibited significantly higher numbers of risk components of MetS. This study also showed that elevated TG was the most prevalent component of MetS. Further investigations are needed to elucidate the determinants of obesity and metabolic syndrome and to emphasize the importance of early screening and preventive interventions.

## Figures and Tables

**Figure 1 ijerph-19-14876-f001:**
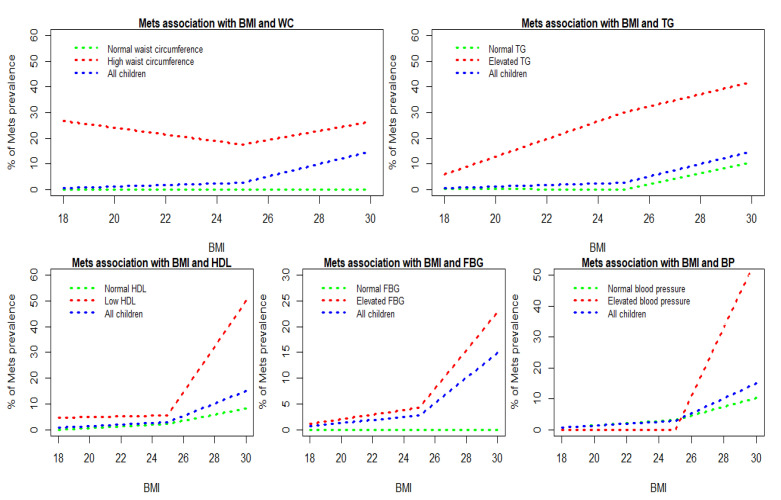
Comparison between metabolic components and BMI among Sudanese early adolescences. Abbreviations: WC—Waist Circumference; BP—Blood Pressure; FBG—Fasting Blood Glucose; TG—Triglycerides; HDL—High-Density Lipoprotein.

**Table 1 ijerph-19-14876-t001:** General background characteristics of the participants based on gender.

Variables	Total (921)	Boys: 388 (42.1%)	Girls: 533 (57.9%)	*p*-Value
Age in years (mean ± SD)	12.59 ± 1.21	12.69 ± 1.16	12.52 ± 1.23	0.027
10 years	57 (6.2%)	20 (5.2%)	37 (6.9%)	
11 years	132 (14.3%)	49 (12.6%)	83 (15.6%)	
12 years	194 (21.1%)	76 (19.6%)	118 (22.1%)	
13 years	295 (32.0%)	131 (33.8%)	164 (30.8%)	
14 years	235 (25.5%)	112 (28.9%)	123 (23.1%)	
15 years	8 (0.9%)	0 (0.0%)	8 (1.5%)	
Type of residence				
Urban	495 (53.7%)	247 (63.7%)	248 (46.5%)	0.001
Rural	426 (46.3%)	141 (36.3%)	285 (53.5%)	
Academic year				
Grade 5	145 (15.7%)	60 (15.5%)	85 (15.9%)	0.050
Grade 6	227 (24.6%)	84 (21.6%)	143 (26.8%)	
Grade 7	224 (24.3%)	88 (22.7%)	136 (25.5%)	
Grade 8	325 (35.3%)	156 (40.2%)	169 (31.7%)	
Parents’ education				
Father’s level of education				
Illiterate	26 (2.8%)	14 (3.6%)	12 (2.3%)	0.416
Khalwa	34 (3.7%)	12 (3.1%)	22 (4.1%)	
Primary	102 (11.1%)	39 (10.1%)	63 (11.8%)	
Intermediate	44 (4.8%)	16 (4.1%)	28 (5.3%)	
Secondary	161 (17.5%)	61 (15.7%)	100 (18.8%)	
University and postgraduate	380 (41.2%)	171 (44.1%)	209 (39.2%)	
Don’t know	174 (18.9%)	75 (19.3%)	99 (18.6%)	
Mother’s level of education				
Illiterate	52 (5.6%)	20 (5.2%)	32 (6.0%)	0.623
Khalwa	33 (3.6%)	16 (4.1%)	17 (3.2%)	
Primary	160 (17.4%)	67 (17.3%)	93 (17.4%)	
Intermediate	52 (5.6%)	19 (4.9%)	33 (6.2%)	
Secondary	217 (23.6%)	83 (21.4%)	134 (25.1%)	
University and postgraduate	302 (32.8%)	135 (34.8%)	167 (31.3%)	
Don’t know	105 (11.4%)	48 (12.4%)	57 (10.7%)	
Weight status				
Underweight	153 (16.6%)	83 (21.4%)	70 (13.1%)	0.004
Normal	573 (62.2%)	231 (59.5%)	342 (64.2%)	
Overweight	108 (11.7%)	36 (9.3%)	72 (13.5%)	
Obesity	87 (9.4%)	38 (9.8%)	49 (9.2%)	

**Table 2 ijerph-19-14876-t002:** Association between prevalence of metabolic syndrome and demographic characteristics.

Variables	Total (921)	Non-MetS Early Adolescents 900 (97.7%)	MetS Early Adolescents 21 (2.3%)	*p*-Value
Gender				
Boys	388 (42.1%)	375 (96.6%)	13 (3.4%)	0.063
Girls	533 (57.9%)	525 (98.5%)	8 (1.5%)	
Age in years				
10 years	57 (6.2%)	56 (98.2%)	1 (1.8%)	0.488
11 years	132 (14.3%)	132 (100.0%)	0 (0.00%)	
12 years	194 (21.1%)	188 (96.9%)	6 (3.1%)	
13 years	295 (32.0%)	288 (97.0%)	7 (3.0%)	
14 years	235 (25.5%)	228 (97.0%)	7 (3.0%)	
15 years	8 (0.9%)	0 (0.00%)	8 (100.0%)	
Type of residence				
Urban	495 (53.7%)	478 (96.6%)	17 (3.4%)	0.011 *
Rural	426 (46.3%)	422 (99.1%)	4 (0.9%)	
Academic year				
Grade 5	145 (15.7%)	144 (99.3%)	1 (0.7%)	0.082
Grade 6	227 (24.6%)	225 (99.1%)	2 (0.9%)	
Grade 7	224 (24.3%)	218 (97.3%)	6 (2.7%)	
Grade 8	325 (35.3%)	900 (97.7%)	21 (2.3%)	
Weight status				
Under weight	153 (16.6%)	152 (99.3%)	1 (0.7%)	0.001 **
Normal weight	573 (62.2%)	569 (99.3%)	4 (0.7%)	
Overweight	108 (11.7%)	105 (97.2%)	3 (2.8%)	
Obese	87 (9.4%)	74 (85.1%)	13 (14.9%)	
Total	921 (100%)	900 (97.7%)	21 (2.3%)	

* Significant, ** Highly Significant.

**Table 3 ijerph-19-14876-t003:** Prevalence of metabolic syndrome components.

Characteristics	Total (921)	Non-MetS Early Adolescents 900 (97.7%)	MetS Early Adolescents 21 (2.3%)	*p*-Value
Waist circumference				
Normal	839 (91.1%)	839 (100.0%)	0 (0.00%)	0.001 **
High	82 (8.9%)	61 (74.4%)	21 (25.6%)	
Systolic blood pressure				
Normal	877 (95.3)	861 (98.2%)	16 (1.8%)	0.001 **
High	43 (4.7)	38 (88.4%)	5 (11.6%)	
Diastolic blood pressure				
Normal	728 (79.1)	716 (98.4%)	12 (1.6%)	0.012 *
High	192 (20.9)	183 (95.3%)	9 (4.7%)	
Blood pressure				
Normal	877 (95.3)	861 (98.2%)	16 (1.8%)	0.001 **
High	43 (4.7)	38 (88.4%)	5 (11.6%)	
Fasting blood glucose				
Normal	313 (34.0%)	313 (100.0%)	0 (0.00%)	0.001 **
High	607 (65.9%)	586 (96.5%)	21 (3.5%)	
Triglycerides				
Normal	856 (93.0%)	845 (98.7%)	11 (1.3%)	0.001 **
High	64 (6.9%)	54 (84.4%)	10 (15.6%)	
HDL-c				
Normal	782 (85.0)	774 (99.0%)	8 (1.0%)	0.001 **
Low	138 (15.0)	125 (90.6%)	13 (9.4%)	

* Significant, ** Highly Significant.

**Table 4 ijerph-19-14876-t004:** Association between metabolic syndrome and its components.

Model	Components	Unstandardized Coefficients (β)	Standardized Coefficients (β)	*p*-Value
1	High WC	0.256	0.489	0.000 **
2	High WC	0.247	0.472	0.000 **
	High TG	0.121	0.207	0.000 **
3	High WC	0.246	0.470	0.000 **
	High TG	0.107	0.182	0.000 **
	Low HDL	0.066	0.158	0.000 **
4	High WC	0.248	0.473	0.000 **
	High TG	0.106	0.180	0.000 **
	Low HDL	0.063	0.150	0.000 **
	High FBG	0.033	0.105	0.000 **
5	High WC	0.244	0.465	0.000 **
	High TG	0.106	0.181	0.000 **
	Low HDL	0.062	0.148	0.000 **
	High FBG	0.031	0.100	0.000 **
	High HTN	0.067	0.095	0.001 **

** Highly Significant.

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
