# Peer review of "Prevalence of Metabolic Syndrome among Early Adolescents in Khartoum State, Sudan"

_ijerph, 2022, doi:10.3390/ijerph192214876_

Round 1

Reviewer 1 Report

The manuscript entitled “Prevalence of metabolic syndrome among early adolescents in Khartoum State, Sudan” by Fatima A Elfaki et al. focuses on the studying of the metabolic syndrome progress. The aim of this study was to estimate the prevalence of Metabolic syndrome and its components among Sudanese early adolescents in Khartoum State. So, the authors of the article found that, boys had significantly higher prevalence of metabolic syndrome than girls. Obese adolescents with high waist circumference exhibited significantly higher numbers of risk components of Metabolic syndrome as compared with the normal or overweight waist circumference early adolescents.

The method used is adequately described and the results obtained are presented well enough. Besides, the conclusions of the authors are supported by the clear graphs.

The results obtained are significant for a better understanding of effective of prevention the pathology in question and can be used for prevent of the development.

The following critical remarks can be made.

1.                 The authors of the article/researchers have not performed the family analysis for the discussion of the hereditary predisposition to pathology.

2.                 It is well known, that slow pace of life causes/can cause/lead to obesity and the development of different pathologies, including cardiovascular diseases. From the results obtained is not clear how often the metabolic syndrome occurred in early adolescents with active lifestyle. 

Author Response

Dear Sir/Madam,

Thank you

Reviewer 2 Report

This manuscript well written. Only the introduction need to be shorter. 

Author Response

Dear Sir/Madam,

Thank you

Reviewer 3 Report

Despite the study is interesting there are several comments author must address.

Authors should describe the novelty of the work more clearly. Discussion should be made in relation of metabolic syndrome parameters (high blood pressure, high blood sugar etc., ) they collected with heart failure and stroke. Authors must provide appropriate reason for the selection of the study site Khartoum State, Sudan. What was the reason behand selecting only two localities (Khartoum and Karari) our of seven studied localities.

The method details on how the High- density lipoprotein cholesterol, serum triglycerides, fasting blood sugar level were measured should be mentioned. The accuracy of the blood samples measurements should be addressed. Explain the statistical tests chi-square test and Multivariable regression function, provide the details including software ver, parameters used for these tests. The reason for measuring blood pressure at 5 min intervals should be included. References are bit old suggest to include latest references and data connected with Sudan or other countries.

Grammatical errors has to be corrected throughout the manuscript.

Author Response

Dear Sir/Madam,

Thank you

Round 2

Reviewer 1 Report

The authors significantly revised the manuscript, taking into account the comments.

Reviewer 3 Report

The authors addressed most of the comments and improved the quality of manuscript.